# Acoustic Vector Sensor Multi-Source Detection Based on Multimodal Fusion

**DOI:** 10.3390/s23031301

**Published:** 2023-01-23

**Authors:** Yang Chen, Guangyuan Zhang, Rui Wang, Hailong Rong, Biao Yang

**Affiliations:** 1School of Microelectronics and Control Engineering, Changzhou University, Changzhou 213159, China; 2College of IoT Engineering, Hohai University, Changzhou 213159, China; 3State Key Laboratory of Automotive Simulation and Control, Jilin University, Changchun 130012, China

**Keywords:** acoustic vector sensor, modal decomposition, density peak clustering, DOA, source counting

## Abstract

The direction of arrival (DOA) and number of sound sources is usually estimated by short-time Fourier transform and the conjugate cross-spectrum. However, the ability of a single AVS to distinguish between multiple sources will decrease as the number of sources increases. To solve this problem, this paper presents a multimodal fusion method based on a single acoustic vector sensor (AVS). First, the output of the AVS is decomposed into multiple modes by intrinsic time-scale decomposition (ITD). The number of sources in each mode decreases after decomposition. Then, the DOAs and source number in each mode are estimated by density peak clustering (DPC). Finally, the density-based spatial clustering of applications with the noise (DBSCAN) algorithm is employed to obtain the final source counting results from the DOAs of all modes. Experiments showed that the multimodal fusion method could significantly improve the ability of a single AVS to distinguish multiple sources when compared to methods without multimodal fusion.

## 1. Introduction

An acoustic vector sensor (AVS) consists of a sound pressure sensor and vibration velocity sensors. It is an acoustic receiving energy converter that can acquire sound pressure and vibration velocity signals and convert underwater acoustic signals into electrical signals. In recent years, acoustic vector sensors have become one of the key technologies in underwater acoustic engineering. The traditional scalar sensor can only pick up the scalar information of acoustic pressure [1]. However, acoustic vector sensors can also pick up vector information such as particle vibration velocity, acceleration, and displacement. A single AVS can be used to estimate the direction of arrival (DOA) of a sound source [2,3,4,5,6], for which it is no longer necessary to use a large hydrophone array. In addition, single AVSs are employed on small, unmanned platforms such as buoys because of their advantages of small size, easy installation, and low cost; this is of great significance to the design of efficient and convenient underwater acoustic systems. The average sound intensity method and the conjugate cross-spectrum method are mainly employed for sound source detection with a single AVS. In the average sound intensity method, the direction is calculated by the arctangent function after multiplying the sound pressure and the vibration velocity. This method has good detection performance under conditions with a high signal-to-noise ratio, but only the synthetic direction can be obtained when faced with multiple sound sources [3]; thus, this method can only be employed for single source detection. The conjugate cross-spectrum method calculates the direction of each time-frequency point by the conjugate cross-spectrum of the sound pressure component and vibration velocity component; then, these DOA estimates are counted to obtain a DOA histogram, from which the direction corresponding to the peak of the cluster is the DOA of the sound sources and the number of the sources can be obtained by counting the number of clusters [7,8,9].

The W-disjoint orthogonality (WDO) is a degree that represents the ability of a DOA histogram based on AVS to distinguish multiple sources [10]. In a single time-frequency point, if only the energy of one source is dominant, whereas that of the other sources is secondary, the time-frequency point is called the dominant time-frequency point of the source. The orientation of this time-frequency point will, therefore, be biased towards the orientation of the source with greater energy. If there are plenty of time-frequency points with such characteristics, these points will gather near the true orientations of the sources to form clusters and present as spectral peaks in the DOA histogram. Due to the WDO of the signals, the number of the sources can be counted as the number of clusters. However, as the number of sound sources increases, the WDO of underwater acoustic signals will be weakened [3], which seriously affects the accuracy of source counting. To solve this problem, intrinsic time-scale decomposition (ITD) can be introduced. ITD is a time-frequency analysis algorithm that can decompose a signal into multiple modes according to the frequency band [11,12,13]. Thus, ITD can be employed to decrease the number of the sources in each mode and increase the WDO of the signal.

In the underdetermined blind source separation problem, source counting is an important issue. Clustering algorithms, which are traditional machine learning technologies [14,15,16], are usually employed to obtain the number of the sources. The *k*-means clustering algorithm (*k*-means) is a commonly employed unsupervised learning algorithm. However, the number of clusters needs to be set in advance [17,18]; therefore, it cannot be employed for source counting. The Gaussian mixture model (GMM) algorithm can achieve high estimation accuracy in underdetermined conditions [19,20]. However, a threshold needs to be set to distinguish between the Gaussian component of the fitted peak and the fitted sidelobe, which requires a series of experiments. When the number of the sources increases, the degree of WDO in underwater acoustic signals becomes weaker, which will affect the difficulty of distinguishing the Gaussian component of the fitted sound source from the Gaussian component of the fitted background noise. By constructing the local density and minimum distance to find cluster centers, density peak clustering (DPC) can not only deal with noise or high-dimensional problems, but can also automatically find cluster centers without setting the number of the sources in advance [21,22].

In order to tackle these problems, this paper proposes a multimodal fusion algorithm based on ITD and density-based spatial clustering of applications with noise (DBSCAN) that is efficient in conditions with multiple underdetermined sound sources. First, the underwater acoustic signal is decomposed into multiple modes according to different frequency bands by the ITD decomposition algorithm to reduce the number of sources in each mode, thereby enhancing the WDO of the signal and improving the detection performance of the AVS. Second, the DOA at each moment is obtained via DPC and the gap-based method [3]. Finally, the DBSCAN algorithm is employed to cluster the source direction samples at each moment, and a final DOA and the source counting results are obtained. Compared with the traditional method, our multimodal fusion algorithm retains the advantages of the ITD and DBSCAN algorithms and improves the accuracy of source counting.

## 2. AVS Multi-Source Detection Algorithm Based on Multimodal Fusion

Figure 1 shows the multimodal fusion algorithm flow. The outputs of the AVS are one-channel sound pressure signals and dual-channel vibration velocity signals. First, ITD decomposition is performed on the three-channel signals collected to obtain multiple modes. Each mode contains one-channel sound pressure signals and dual-channel vibration velocity signals corresponding to the output of the AVS. After the underwater acoustic signal is decomposed by ITD, the number of sources decreases in each mode, which will increase the WDO of the underwater acoustic signals. Second, the number and orientation of the sources in each mode can be obtained by employing DPC and the gap-based method after calculating the conjugate cross-spectrum of each mode. Finally, the DBSCAN algorithm is employed to fuse the orientation samples of selected modes to obtain the number of sources and the DOA.

### 2.1. Intrinsic Time-Scale Decomposition

The WDO of underwater acoustic signals will decrease as the number of sources increases, at which point the ability of a single AVS to distinguish multiple sources will decrease [10]. If the number of sources can be reduced, the accuracy of underdetermined source counting will be improved. The signals that radiate from sources have different frequency bands; thus, we can correctly distinguish different sources in underdetermined conditions using ITD.

The ITD algorithm can be used to decompose a non-stationary signal into a set of proper rotation components (PRCs) and a monotonic trend signal according to frequency bands [23,24,25]. Each PRC is uncorrelated in the frequency domain. The main steps of ITD decomposition are as follows:

For an original signal $X_{t}$,$\tau_{k}$ (*k* = 1, 2…) is its local extreme point, and $L_{t}$ and $H_{t}$ represent the PRC and monotonic trend signal, respectively. When determining a linear function between two extreme points $\tau_{k}$ and $\tau_{k+2}$, the value of this function at the extreme point $\tau_{k+1}$ can be expressed as:(1)$$S_{k+1}=X_{k}+\frac{\tau_{k+1}-\tau_{k}}{\tau_{k+2}-\tau_{k}}(X_{k+2}-X_{k})$$

The value of $L_{t}$ on the extreme point $\tau_{k+1}$ can be defined as:(2)$$L_{k+1}=aS_{k+1}+\left(1-a\right)X_{k+1}$$
where a∈[0,1], which determines the amplitude of $L_{t}$. Here, we define a baseline extraction operator ξ [22], such that the value between extreme point $\tau_{k}$ and extreme point $\tau_{k+1}$ can be expressed as a linear transformation:(3)$$L_{t}=\xi X_{t}=L_{k}+\frac{L_{k+1}-L_{k}}{X_{k+1}-X_{k}}(X_{t}-X_{k})$$

If $L_{t}$ is not a monotonic function, it needs to be decomposed by ITD continually. Every time it is decomposed, a new rotation component and baseline signal are obtained. The original signal can be expressed as:(4)$$\begin{array}{ll} X_{t} & =HX_{t}+\xi X_{t}\\ & =HX_{t}+\xi (H+\xi )X_{t}\\ & =(H\sum_{m=0}^{p-1}\xi^{m}+\xi^{p})X_{t} \end{array}$$
where H+ξ=1, $H\xi^{m}X_{t}$ is the PRC obtained by the *m* + 1th ITD decomposition and $\xi^{p}X_{t}$ is a monotonic trend component.

After the underwater acoustic signal is decomposed by ITD, each PRC is a mode, and the frequency band of each mode decreases as the number of decompositions increases; different sources have different frequency bands. Therefore, after ITD decomposition, the number of sources in each mode will be less than that in the original signal. This can improve the WDO of underwater acoustic signals, and the ability of a single AVS to distinguish between multiple sources in underdetermined conditions will also be improved. If the frequency bands are considerably different between each source, it is possible to directly separate them [3].

### 2.2. DPC with the Gap-Based Method

After ITD decomposition, DPC and the gap-based method [26,27,28] is employed to cluster the DOA estimates of all time-frequency points and count the source number of each mode. The DOA estimates of each time-frequency point can be obtained by the conjugate cross-spectrum of the sound pressure and vibration velocity of each mode. The model of a two-dimensional AVS can be expressed as:(5)$$\{\begin{array}{c} p_{i}(t)=x_{i}(t)+n_{pi}(t)\\ v_{xi}(t)=x_{i}(t)\cos\alpha +n_{xi}(t)\\ v_{yi}(t)=x_{i}(t)sin\alpha +n_{yi}(t) \end{array}$$
where $x_{i}(t)$ is the sound source signal of the ith mode; $p_{i}(t)$, $v_{xi}(t)$ and $v_{yi}(t)$ are the sound pressure signal and the vibration velocity signal of the *i*th mode in two directions, respectively, collected by the AVS; $n_{pi}(t)$,$n_{xi}(t)$ and $n_{yi}(t)$ are the noise of the ith mode; and α is the angle between the sound source and the AVS.

The three channels are together converted to a time-frequency domain by short-time Fourier transform (STFT); then, the conjugate cross-spectrum of the sound pressure and vibration velocity can be employed to obtain the direction of the source:(6)θi(ω,m)=tan−1[Re{Pi(ω,m)×Vyi*(ω,m)}Re{Pi(ω,m)×Vxi*(ω,m)}]
where $P_{i}(\omega ,m)$, $V_{xi}(\omega ,m)$ and $V_{yi}(\omega ,m)$ are the STFT of $p_{i}(t)$, $v_{xi}(t)$ and $v_{yi}(t)$, respectively, and $V_{xi}^{*}(\omega ,m)$ and $V_{yi}^{*}(\omega ,m)$ represent the complex conjugates of $V_{xi}(\omega ,m)$ and $V_{yi}(\omega ,m)$, respectively.

By employing DPC and the gap-based method for each time-frequency point, clusters of the DOA estimates can be obtained, where the number of clustering centers is the source number and the DOA corresponding to the cluster center is the DOA of the source. The algorithm first calculates the local density and minimum distance of each bearing sample. Then, the local density and the minimum distance are multiplied and arranged in reverse order. The variance in the difference is calculated after the difference in the product is obtained:(7)$$\sigma_{n}^{2}=\frac{1}{K-n}{\sum_{i=n}^{K-1}\left(\Delta \gamma_{i}-\frac{1}{K-n}\sum_{i=n}^{K-1}\Delta \gamma_{i}\right)}^{2}$$
where $\gamma_{i}$ is the product of local density and minimum distance, $\sigma_{n}^{2}$ is the variance of $\gamma_{i}$, *K* is the total number of bearing samples, and *n* is the sample number.

The number of sources can be obtained from Equation (7):(8)$$N=\underset{n=1,\ldots ,K-3}{\arg\min}(\frac{\sigma_{n+1}^{2}}{\sigma_{n}^{2}})$$

The direction corresponding to the first *N* samples with the largest variance is the clustering center. By employing DPC with the gap-based method, the number and DOA of the sources in each mode are obtained. However, the DOAs of the same source in different modes are close, but not identical, and the DOAs of all modes need to be clustered. DPC cannot distinguish the directions of these modes because there are few direction samples.

### 2.3. Multimodal Fusion

Through ITD decomposition and conjugate cross-spectral orientation estimation, we can obtain the number and DOA of the sources in each mode; however, due to the reduction in the number of sources in each mode, the obtained number and DOA of sources are not the true values. In addition, there may be irrelevant DOAs in these modes. Therefore, these directions cannot be directly merged. Thus, DBSCAN is employed to cluster the DOA of the source in each mode and obtain a final DOA and number of sources.

The DBSCAN algorithm examines the connectivity between samples from the perspective of sample density and continuously expands clustering based on connectivity to obtain the final clustering results. The DOAs of the same source are very close in each mode; thus, the DOAs of the same source can be merged by using the connectivity judgment of DBSCAN.

The core concept of DBSCAN is to find dense regions of sample points, which can be viewed as a cluster [29]. The algorithm has the following definitions:

EPS-neighborhood: ∃ aggregation $D=\{x_{1},x_{2}\ldots x_{m}\}$, for $\forall x_{j}\in D$, EPS-neighborhood represents all the sample points in the circle with $x_{j}$ as the center and EPS as the radius.

Core object: object with at least M objects within a radius of ‘Eps-neighborhood’.

Directly density-reachable: if point $x_{j}$ is in the EPS-neighborhood of point $x_{i}$ and point $x_{i}$ is the core object, then point $x_{j}$ is directly density-reachable from point $x_{i}$.

Density-reachable: for points $x_{i}$ and $x_{j}$, if ∃ is a set of points $p_{1},p_{2}\ldots p_{n}$ and point $p_{i+1}$ is directly density-reachable from point $p_{i}$, where $p_{1}$ = $x_{i}$ and $p_{n}$ = $x_{j}$, then point $x_{j}$ is density-reachable from point $x_{i}$.

As shown in Figure 2, the DBSCAN algorithm first takes any core object as the starting point and finds all the sample points that are density-reachable from that point: these points form the first cluster. Then, these steps are repeated for the remaining sample points. After all the sample points are selected, the number of clusters can be obtained. Thus, by employing DBSCAN to fuse the orientation samples of selected modes, DOAs belonging to the same source are clustered, and the number of sources can be obtained.

Due to the fact that the DBSCAN algorithm is very sensitive to the selection of parameters (EPS, M), this paper introduces a modified DBSCAN algorithm that can determine these parameters adaptively [30]. Firstly, the distance matrix of all sample points is sorted by row, and the mean value of each column of the new matrix is obtained as the value of EPS. Then, for each EPS, the corresponding M value is obtained, where M is the mean number of samples in the EPS-neighborhood of all sample points. Finally, DBSCAN is performed with the EPS and M value of each group as the parameters, and the values of the group with the smallest change in the number of clusters are taken as the final DBSCAN parameters. After the DOA estimates of each mode are clustered by DBSCAN, the DOA estimates unrelated to the source are eliminated, and the actual DOAs of the sources are retained. Finally, a final DOA and number of sources and can be obtained.

## 3. Experimental Setup

We used noise radiated by boats collected in Fuxian Lake to verify the algorithm. As shown in Figure 3, the AVS was placed in the middle of the lake. There were four small sailing boats in uniform motion around the perimeter, which was 1–2 km from the AVS. The initial directions of the boats were uniformly distributed within four degrees of 40°, 140°, 210°, and 320° from the AVS. In the experiment, a two-dimensional co-vibration vector hydrophone was employed for signal sampling. The sensitivity of the velocity channel decreased with increasing frequency in the slope of the −6 dB/octave, and the phase difference between the two channels of sound pressure and vibration velocity was 90 degrees. The output of the AVS was collected with a multi-channel synchronous data collector.

To improve the accuracy of the DOA and source counting, the original signals were regarded as a mode in the modal fusion stage, which makes the cluster more compact and reduces the number of discrete points. The other parameters of the experiment were as follows: the weight a of ITD decomposition was 0.5, the sampling rate was 48 kHz, the working frequency band of AVS was 5–8 kHz, the STFT window length was 8192 sampling points, and the sliding step length was 4096 sampling points. There were two sets of experiments. The first group of experiments used the multimodal fusion algorithm to calculate the DOA and number of sources in each time frame. In the second set of experiments, the DOA and number of sources were calculated without using the multimodal fusion algorithm. The results of the two groups of experiments were compared and analyzed.

## 4. Results and Discussions

Figure 4 shows the traces of the boats in the first four modes obtained using the multimodal fusion algorithm. It can be seen that the number of sound sources in each mode was different. As the number of decompositions increased, the number of the sources decreased. Because different sources occupy different frequency bands, the more dominant time-frequency points a sound source has, the clearer its trace is. In addition, the traces of the sources in Figure 4 were found to break at some time frames as a result of too few dominant time-frequency points for these sources at these time frames. If a source does not have enough dominant time-frequency points, the DOA estimates corresponding to the source are unable to form clusters. However, the missing parts of these traces can be found in other modes. For instance, the trace of the third source in mode 1 breaks within 18–22 s in Figure 4a, but it can be clearly observed in the third mode in Figure 4c. Similarly, the missing part of the two targets in mode 4 within 30–35 s in Figure 4d can be observed in mode 3 in Figure 4c.

The reason for this phenomenon is that different sources occupy different frequency bands. The number of dominant time-frequency points of one source is directly proportional to its bandwidth. If a source has sufficient dominant time-frequency points, these time-frequency points will aggregate to form clusters according to the energy of the dominant source, where the direction corresponding to the cluster center is the direction of the sound source. Since the number of time-frequency points is certain, more dominant time-frequency points in one source means less dominant time-frequency points in other sources, and sound sources with few dominant time-frequency points are difficult to detect. After modal decomposition, a sound source that is dominant in one mode may be secondary in other modes at the same time frame. Simply put, other sound sources will have more dominant time-frequency points in these modes to form clusters, and their traces can become more continuous. As long as the DBSCAN algorithm is performed on the directions of all modes, a final number of sources and their traces can be obtained.

Figure 5a–c show the traces of the boats estimated by the multimodal fusion algorithm. Compared with the traces obtained without employing the multimodal fusion algorithm in Figure 5d, it can be clearly seen that the traces obtained by the multimodal fusion algorithm are more continuous: the four boats start from 40°, 140°, 210°, and 320° from the AVS and arrive at 0°, 80°, 200°, and 280°, respectively, fifty seconds later. The multimodal fusion algorithm can obtain orientations which could not have been obtained due to interference from other sources. The traces of the boats obtained without using multimodal fusion shown in Figure 5d are not continuous because of the interference of other boats, especially when the traces of the first two boats are between 10 and 20 s and the trace of the third boat is between 40 and 50 s. In Figure 5b,c, the traces of four boats are more continuous, and the directions that disappear due to the interference of other boats in Figure 5d can be observed clearly. Moreover, with an increase in the mode number employed, the traces become more complete. When employing four modes of data, the traces of the four boats are almost completely continuous.

Figure 6a–c show the quantitative distribution of sound sources obtained by employing two, three, and four modes, respectively. Figure 6d shows the quantitative distribution of sound sources obtained without using the multimodal fusion algorithm. In this experiment, the true number of sources was four. For the process employing two modes, shown in Figure 6a, the quantitative distribution of the sources is not ideal due to imperfect source direction information contained in these modes. However, with an increase in the number of modes used in the fusion, the histogram of the quantitative distribution of the source numbers becomes more accurate.

In Figure 6b,c, it can be observed that for the group of experiments employing the multimodal fusion algorithm, the number of sources was equal to the true value 43% and 46% of the time, respectively, while those that did not employ the modal fusion algorithm accounted for only 32%. After multimodal fusion, the accuracy of source counting increased by 11% and 14%. From the trend in the quantitative distribution of source numbers, the proportion of the source number after employing the multimodal fusion algorithm increases with the increase in the number of the sources; the maximal value is obtained when the source number is four, then the proportion of the sources decreases gradually. The closer to the true value of the source number, the higher the proportion of the source numbers, and the more the quantitative distribution of the source number obeys the Gaussian distribution.

## 5. Conclusions

In this study, ITD decomposition and DBSCAN were combined for underdetermined source counting. Under multi-source conditions, the ability of a single AVS to distinguish multiple sources will decay. ITD decomposition was employed to decompose the signal into multiple modes, decreasing the number of sources in each mode and improving the accuracy of DOA. On this basis, DBSCAN was employed to cluster the source directions in each mode in order to obtain final traces and the number of sources. The experimental results showed that the performance of the multimodal fusion algorithm was better than that of methods without multimodal fusion, whether employing the orientation information of three or four modes. With respect to the traces of the sources, the multimodal fusion algorithm can obtain orientations which could not have been obtained due to interference from other sources. The traces obtained using the multimodal fusion algorithm had fewer discontinuity points and were more continuous. From the source counting results, the proportion of four sources was improved to varying degrees compared with the method without multimodal fusion. The closer to the true value of the estimated source number, the higher the proportion of the number, and the number of sources corresponding to the peak was the true value of source number. The ability of a single AVS to distinguish multiple sources was, therefore, improved by the multimodal fusion algorithm.

## Figures and Tables

**Figure 1 sensors-23-01301-f001:**
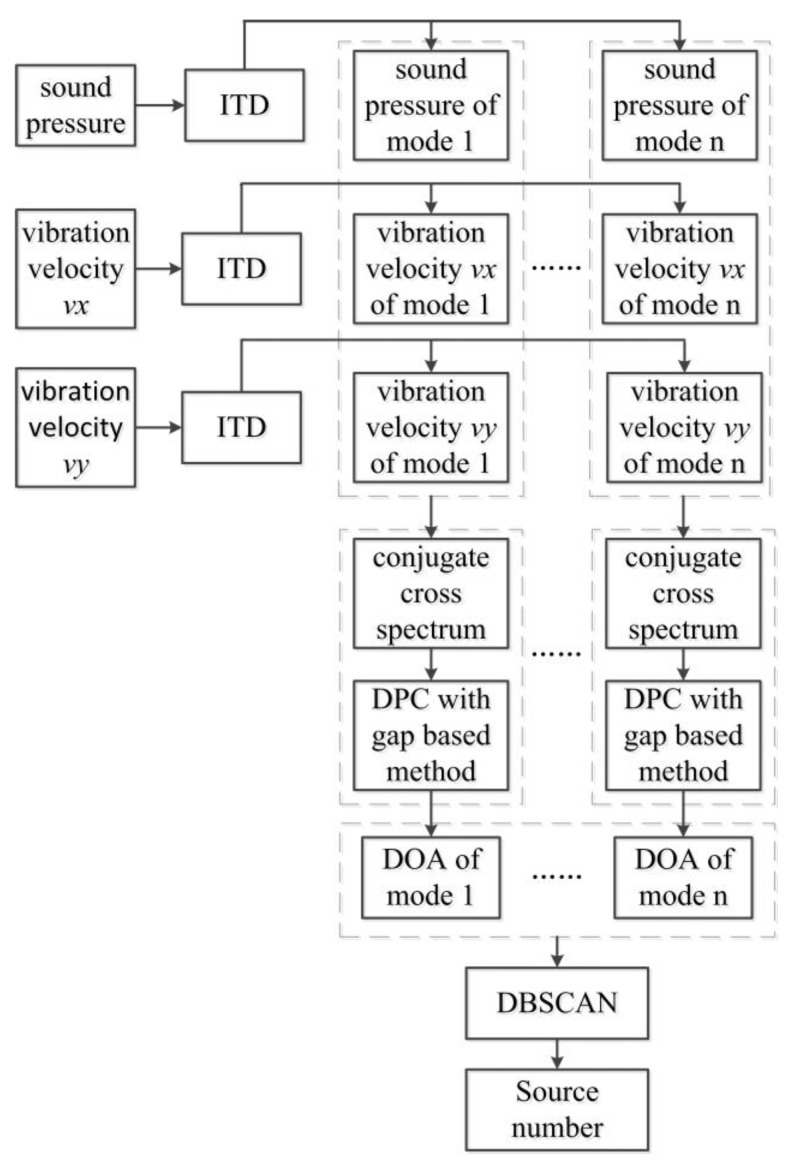
Flow chart of multimodal fusion.

**Figure 2 sensors-23-01301-f002:**
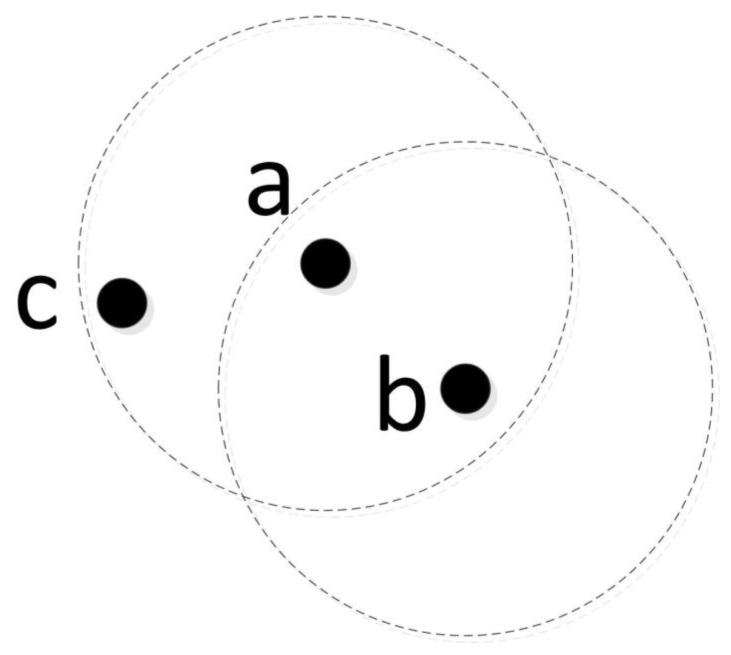
Definition of density reachability; both point *a* and point *b* are core objects, where point *a* is in the EPS-neighborhood of point *b*, and point *c* is in the EPS-neighborhood of point *a*, but not in the EPS-neighborhood of point *b*. It can be observed from the above definition that point *c* is directly density-reachable from point *a* and point *c* is density-reachable from point *b*; thus, point *a*, point *b*, and point *c* can be classified into the same category.

**Figure 3 sensors-23-01301-f003:**
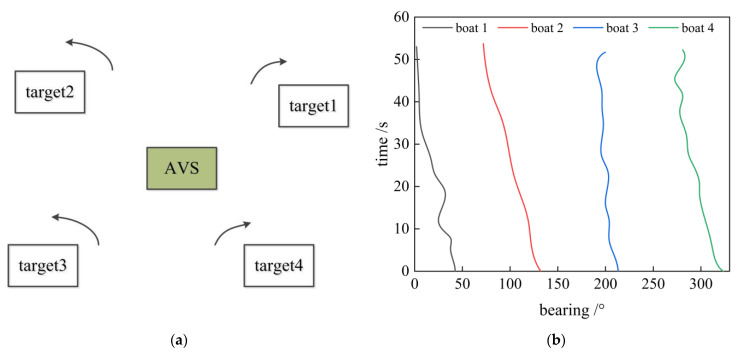
Experimental setup and traces of four boats: (**a**) experimental setup; (**b**) experiment azimuth waterfall sketch map.

**Figure 4 sensors-23-01301-f004:**
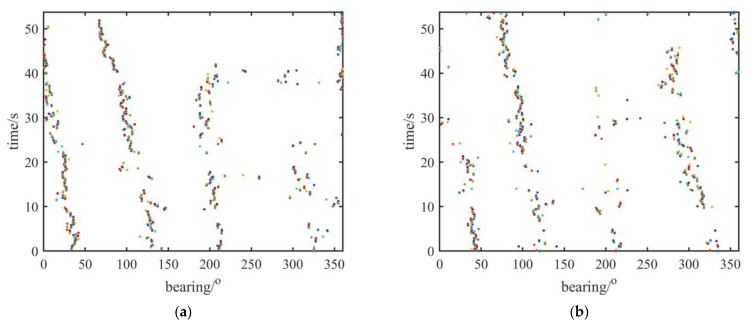
Traces of the boats in the first–fourth mode: (**a**) traces of boats in mode 1; (**b**) traces of boats in mode 2; (**c**) traces of boats in mode 3; (**d**) traces of boats in mode 4.

**Figure 5 sensors-23-01301-f005:**
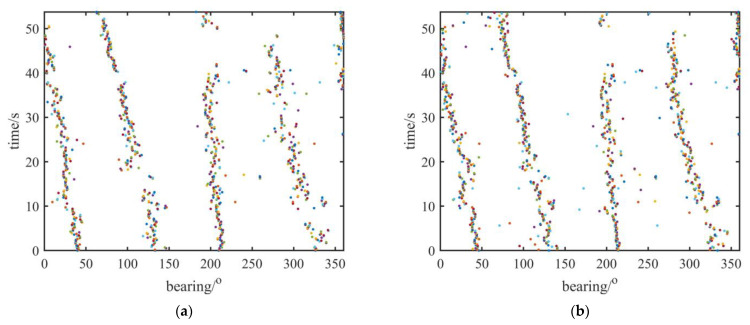
Traces of boats estimated by multimodal fusion algorithm: (**a**) traces of boats employing two modes, including original signal and mode 1; (**b**) traces of boats employing three modes, including original signal, mode 1, and mode 2; (**c**) traces of boats employing four modes, including original signal, mode 1, mode 2, and mode 3; (**d**) traces of boats obtained without using multimodal fusion.

**Figure 6 sensors-23-01301-f006:**
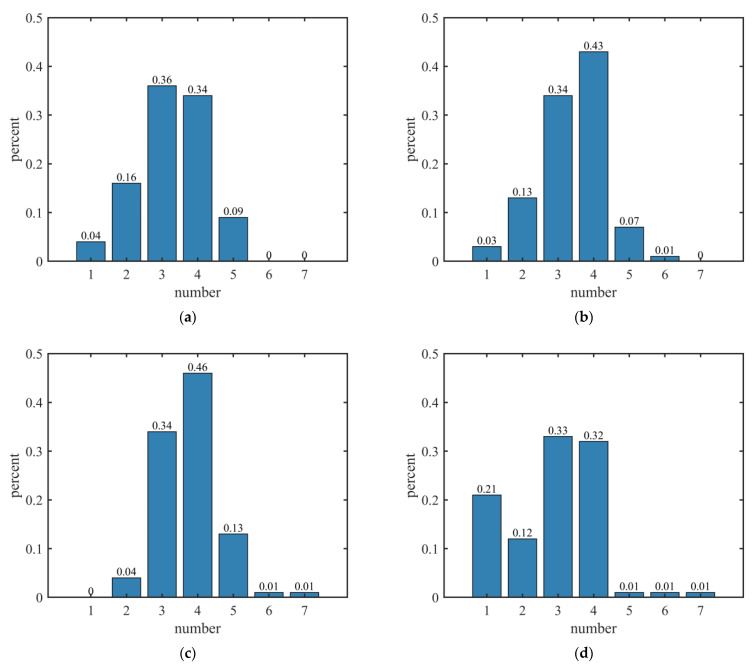
Quantitative distribution of source number: (**a**) quantitative distribution of source number employing two modes, including original signal and mode 1; (**b**) quantitative distribution of source number employing three modes, including original signal, mode 1, and mode 2; (**c**) quantitative distribution of source number employing four modes, including original signal, mode 1, mode 2, and mode 3; (**d**) quantitative distribution of source number without multimodal fusion.

## Data Availability

Not applicable.

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
