# Peer review of "Acoustic Vector Sensor Multi-Source Detection Based on Multimodal Fusion"

_sensors, 2023, doi:10.3390/s23031301_

Round 1
Reviewer 1 Report
In this paper, it proposes a multi-sources detection method for acoustic vector sensor based on the multi-modal fusion way. The idea is basically new and the work is somewhat valuable.
However, the manuscript is not prepared well and there are a lot of typo-errors in it.
1. How to use the different source number estimate to obtain the final correct source number? It is not clear. Especially, why and how to use the DBSCAN to solve this problem? The basic solution way is not so clear.
2. There are a lot abbreviations, which are confusing. Some of them are not defined properly when it first appeared. Please check the whole manuscript and refine it.
3. There are many errors in Fig. 1. The vibration velocity vy is mistaken as vx. And some "Mode n" is mistaken as "Mode 1".
4. There are a lot of English writing errors. For example, in Line 203, AS ->As; Line 205, three degrees -> four degrees; Line 213, are regard -> are regarded; line 231, are Inability...?
The next version of the manuscript needs to be modified and updated carefully.
Author Response
Dear reviewer
Thank you very much for your suggestions and comments, and my answer is as follows:
1 question: How to use the different source number estimate to obtain the final correct source number? It is not clear. Especially, why and how to use the DBSCAN to solve this problem? The basic solution way is not so clear.
answer: Original signal are divided into into multiple modes,and we can obtain the orientation of each mode,but he the DOAs of the same source in different modes are close but not identical, we need to merge the DOAs of the same target. Thus, we can use the connectivity of DBSCAN to group orientation samples which are close to each other into one class. The number of sources is also obtained. I have added some paragraph to explain why and how use DBSCAN to obtain the final correct source number。
2 question:There are a lot abbreviations, which are confusing. Some of them are not defined properly when it first appeared. Please check the whole manuscript and refine it.
answer: All the abbreviations has defined when it first appeared in the introduction. I have deleted an unnecessary abbreviation。
3 question: There are many errors in Fig. 1. The vibration velocity vy is mistaken as vx. And some "Mode n" is mistaken as "Mode 1"
answer: Error in flowchart has been corrected
4 question: There are a lot of English writing errors. For example, in Line 203, AS ->As; Line 205, three degrees -> four degrees; Line 213, are regard -> are regarded; line 231, are Inability...?
answer: This paper has undergone extensive English revisions, some word and grammar mistakes are corrected.
More details about the article modification are in the attachment.
Yours sincerely
Guanyuan Zhang

Reviewer 2 Report
This paper presents an algorithm based on the combination of the multimode decomposition and clustering of the multiple sources. The authors claim that such a combination allows improving both source counting and the DOA detection. The proposed combined algorithm could have a potential interest from researchers. However, there are questions about the presented work that the authors could hopefully address:
- The authors must improve the introduction part, especially by introducing all the abbreviations.
- Fig. 2 is confusing. Consider adding indices to differentiate different modes. Also, there is a typo regarding vy. Moreover, similar typos can be found in the rest of the text.
- By looking at the Fig.5, I do not see big differences in the performance. Without giving details, it is difficult to notice improvements. Perhaps the authors can describe it in more detail.
- The latter argument puts the performance of the method in question. It is better at getting the number of sources. However, what about the DOA? A simple metric would be appreciated and will highlight better the advantages of the proposed method. Kindly explain what this reviewer missed.
Author Response
Dear reviewer
Thank you very much for your suggestions and comments, and my answer is as follows:
1 question: The authors must improve the introduction part, especially by introducing all the abbreviations.
answer: This paper has undergone extensive English revisions. All the abbreviations has defined when it first appeared in the introduction. Delete an unnecessary abbreviation。
2 question:Fig. 2 is confusing. Consider adding indices to differentiate different modes. Also, there is a typo regarding vy. Moreover, similar typos can be found in the rest of the text.
answer: Fig. 2 is to help understand how DBSCAN clusters similar sample and I have added figure legends to provide more explanation. I have employed mode 1 to replace the first mode,for scenarios using multiple modalities,I clearly explains which modes are used. About typo regarding,I have corrected it.
3 question: By looking at the Fig.5, I do not see big differences in the performance. Without giving details, it is difficult to notice improvements. Perhaps the authors can describe it in more detail.
answer: I have added a paragraph to give more detail to describe improvements of DOAs.
4 question: The latter argument puts the performance of the method in question. It is better at getting the number of sources. However, what about the DOA? A simple metric would be appreciated and will highlight better the advantages of the proposed method. Kindly explain what this reviewer missed.
answer: For the number of sources, this method can improve the percent of correct count of the number of sources, and for the DOA, it can obtain orientations which could not have been obtained due to interference from other sources,thus the traces of sources become more continuous. A experiment azimuth waterfall sketch map is added to measure the improvement of DOA.
More details about the article modification are in the attachment.
Yours sincerely
Gungyuan Zhang

Round 2
Reviewer 2 Report
I thank the authors for the effort to improve their manuscript. I suggest modifying figures 3-5 to overlap the experimental sketch map with the results. That would significantly increase the visibility of the method.
Author Response
Dear reviewer:
I am appreciate a lot for your review work and suggestions. For your suggestions, my modifications are as follows :
Question: I suggest modifying figures 3-5 to overlap the experimental sketch map with the results. That would significantly increase the visibility of the method.
Answer: I have modified Figure 3, now it can more directly reflect the trace of the four boats. Figure3(b) is sketch map of trace of boats. Figure 5 is the orientation of multiple sources at different moments, the directions of all moments form traces. However, source number varies in different moments. The number of sources at a certain moment may be two, three or four. It hard to distinguish which source a certain direction belongs to and it is difficult to fit a curve. So I employ the form of scatter plot to reflect.
Yours sincerely
Mr. Guangyuan Zhang
